# Evaluation of Microbial Dynamics of Kombucha Consortia upon Continuous Backslopping in Coffee and Orange Juice

**DOI:** 10.3390/foods12193545

**Published:** 2023-09-24

**Authors:** Maret Andreson, Jekaterina Kazantseva, Esther Malv, Rain Kuldjärv, Reimo Priidik, Mary-Liis Kütt

**Affiliations:** 1Center of Food and Fermentation Technologies, Mäealuse 2/4, 12618 Tallinn, Estonia; maret@tftak.eu (M.A.); esther@tftak.eu (E.M.); rain@tftak.eu (R.K.); reimo@tftak.eu (R.P.); maryliis@aiotech.bio (M.-L.K.); 2Department of Chemistry and Biotechnology, School of Science, Tallinn University of Technology, Ehitajate Tee 5, 19086 Tallinn, Estonia

**Keywords:** kombucha, backslopping, lactic acid bacteria, metagenomic analysis, fermented beverage, coffee, orange juice

## Abstract

The kombucha market is diverse, and competitors constantly test new components and flavours to satisfy customers’ expectations. Replacing the original brewing base, adding flavours, or using “backslopping” influence the composition of the symbiotic starter culture of bacteria and yeast (SCOBY). Yet, deep characterisation of microbial and chemical changes in kombucha consortia in coffee and orange juice during backslopping has not been implemented. This study aimed to develop new kombucha beverages in less-conventional matrices and characterise their microbiota. We studied the chemical properties and microbial growth dynamics of lactic-acid-bacteria-tailored (LAB-tailored) kombucha culture by 16S rRNA next-generation sequencing in coffee and orange juice during a backslopping process that spanned five cycles, each lasting two to four days. The backslopping changed the culture composition and accelerated the fermentation. This study gives an overview of the pros and cons of backslopping technology for the production of kombucha-based beverages. Based on research conducted using two different media, this work provides valuable information regarding the aspects to consider when using the backslopping method to produce novel kombucha drinks, as well as identifying the main drawbacks that need to be addressed.

## 1. Introduction

Kombucha is the common name for an ancient tea-based beverage fermented with a symbiotic community of acetic acid bacteria (AAB) and yeast. The symbiotic culture of bacteria and yeast in kombucha is named SCOBY. A variety of kombucha species have been identified so far. The most commonly detected bacterial species belong to the genera *Komagataeibacter* and *Gluconobacter* [1,2]. However, genera of lactic acid bacteria (LAB) *Lactobacillus* and *Oenococcus* have also been found in kombucha [2,3]. The identified yeasts generally belong to the genera *Saccharomyces, Dekkera/Brettanomyces, Zygosaccharomyces, Pichia, Torulaspora*, *Candida* and *Schizosaccharomyces* [2,4]. The kombucha fermentation process occurs spontaneously and changes rapidly due to the cooperative and competitive processes required for nutrient symbiosis of bacteria and yeast [5]. Each cultivation leads to the formation of a new SCOBY layer that can be used to start new fermentation [6]. The addition of a small portion of the previous batch of fermented food or drink to refresh or start a new fermentation stage is named “backslopping” (BS) [7]. This method is well established by homebrewers to renew the kombucha culture and has been used for centuries.

Traditional kombucha is made of black or green tea, and usually has 5–10% of added sugar. The kombucha culture has been used to ferment different juices, such as pomegranate, red grape, sour cherry, and apple juice [8]. Sun et al. [9] used kombucha culture to ferment wheatgrass juice to accelerate the increase of antioxidant liberation, such as those in phenolic compounds and several flavonoids. Nizioł-łukaszewska et al. [10] investigated green coffee bean fermentation with a kombucha culture. They detected that the fermented beverage had biologically active compounds which were potential candidates for antioxidants. Additionally, Tu et al. [11]) used kombucha culture to ferment a by-product originating from the production of tofu and soy protein isolates. The authors showed an enhancement of antioxidant activity and greater antimicrobial activity of fermented soy whey. Another part of kombucha studies has focused on the dynamical changes of culture during one growth cycle in different fermentation times [6,12]. Savary et al. [13] created a kombucha culture from the AAB and yeast species previously selected by Coton et al. [4] and monitored the microbial, chemical and physical changes during one batch cycle [4,13].

The origin and composition of kombucha culture and growth matrix and the number of backslopping cycles influence the chemical and microbial properties of the final product [5,14]. Due to the growing popularity of commercial kombuchas, producers are finding ways to introduce new flavours and nuances for this traditional drink. One of the ways is to modify the SCOBY by introducing new bacterial cultures that affect the sensory properties of the drink. Another option is to replace the classical growth environment, usually black or green tea, with a new one. However, it is important to maintain the constant quality of every batch of the final product and obtain stable and reproducible results, which could be possible with the application of backslopping technology. An examination of the effects of SCOBY modification and a profound description of chemical and microbial changes in kombucha community in different novel growth media during backslopping has not been implemented yet. Therefore, we wanted to characterise modified kombucha fermentation in new environments using backslopping methodology with continuous chemical and microbiological monitoring throughout. Based on the research in two different media, orange juice and coffee infusion, in this work we provide valuable information regarding the aspects to consider for this type of kombucha production and what are the main drawbacks to be solved.

## 2. Materials and methods

### 2.1. Creation of Kombucha Culture

The initial kombucha consortium, originating from a commercial Estonian kombucha brewery, was re-inoculated into a 10% sucrose black-tea mixture. The fermentation continued for seven days at 30 °C, after which the lactic acid bacteria (LAB) species were added to the initial kombucha culture. The LAB strains were *Levilactobacillus brevis*, *Lactiplantibacillus plantarum*, *Pediococcus pentosaceus* and *Companilactobacillus paralimentarius*, and they were obtained from the TFTAK’s (Center of Food and Fermentation Technologies, Tallinn, Estonia) culture collection. The strains were originally isolated from sourdough [15]. The LAB species were pre-cultured in a de Man, Rogosa, Sharpe (MRS) broth (Lab M, Neogen Company, London, UK) at 30 °C for 12 h. Based on the optical density (OD), some strain biomass was concentrated, so that all strains had the same initial OD value before introduction into the kombucha. All strains were added in equal amounts at the concentration rate of 1% (1 × 10^7^ cells/mL). The kombucha fermentation supplemented with LAB species continued for two weeks.

### 2.2. Kombucha Cultivation in Coffee and Orange Juice

The experimental plan of the study is shown in Figure 1. The coffee infusion was made using ground coffee beans (100% Arabic) purchased from a local market and boiled in tap water. The coffee concentration in the infusion was 51 g/L, supplemented with 10% of sucrose. The coffee infusion was brewed for 3 min and then filtered through a paper coffee filter into a beaker and cooled down to 25 °C. The pasteurised orange juice was purchased from a local market.

The LAB-tailored kombucha culture was inoculated in orange juice and coffee infusion. Kombucha SCOBY was divided into eight equal sectors, and one sector was added to the fresh matrix (either in orange juice or in coffee). As the initial pH of the orange juice was 4, a piece of SCOBY was added to the orange juice without pH adjustment. In the case of the coffee environment, at first, the pH was adjusted to 4.5 with the initial batch of kombucha liquid, and thereupon, a piece of SCOBY was added. The backslopping experiment continued for five backslopping cycles. The duration of a single cycle varied between the environments: two days for the orange-juice kombucha and four days for the coffee kombucha.

### 2.3. Determination of pH and Titratable Acidity

Five millilitres of kombucha sample were suspended in 50 mL of distilled water to measure pH and titratable acidity (TA). For the pH and TA analyses, the Food and Beverage Analyzer (Mettler-Toledo International Inc., Columbus, OH, USA) was used. The acidity was determined through titration to pH 7 using 0.1 N of NaOH and expressed as 0.1 N of NaOH per 1 mL of kombucha. All measurements were performed in sets of two replicate measurements.

### 2.4. Determination of Metabolic Products in Kombucha

13 mm Philic PTFE 0.2 μm Non-sterile Millex-LG filters (Millipore, Darmstadt, Germany) were used to filter the kombucha samples. A Waters 2695 HPLC system (Waters Corporation, Milford, MA, USA) was used to analyse the metabolic products (organic acids, sugars, and ethanol) of the kombuchas. An HPX-87H column (BIO-RAD Hercules, Hercules, CA, USA) was used, and the system was eluted isocratically with 0.005 M of H_2_SO_4_ at 0.6 mL/min at 35 °C. A Waters 2487 dual absorbance detector (Waters Corporation, Milford, MA, USA) and a Waters 2414 refractive index detector (Waters Corporation, Milford, MA, USA) were used for the detection and quantification of analytes. Empower software 3 (Build 3471 FR5 SR4, Waters Corporation, Milford, MA, USA) was used to perform the data analysis.

### 2.5. Microbial Quantitative Metagenomic analysis

#### 2.5.1. Sample Preparation

For the microbial analysis of the original, laboratory-modified by lactic acid bacteria (LAB-tailored), and backslopping kombucha, 40 mL of liquid and 0.75 mL of SCOBY samples were collected aseptically. During the backslopping, the samples were taken according to the experimental plan (Figure 1)—at the starting point from tailored kombucha (BS 0.0) and at each end of the backslopping cycle from orange juice (BS 0.2, 1.2, 2.2, 3.2, 4.2, 5.2) and coffee kombucha (BS 0.4, 1.4, 2.4, 3.4, 4.4, 5.4). In the case of the orange-juice kombucha, only the sample of SCOBY was taken. The liquid sample was prepared as described in Andreson et al. [2], with additional centrifugation at the beginning at 700× *g* for 5 min at 4 °C. The supernatant was poured into a new tube and centrifuged at 3950× *g* for 15 min at 4 °C. The SCOBY sample was cut into small pieces with a sterile scalpel and then washed with 750 µL of sterile and cold 1 × PBS (Phosphate-buffered saline, BIO-RAD, Hercules, CA, USA). The sample was centrifuged at 10,000× *g* for 10 min at 4 °C, and the pellet was subjected to the gDNA extraction. The gDNA isolation of each type of kombucha sample was performed and assessed as in Andreson et al. [2].

#### 2.5.2. Next-Generation Amplicon Sequencing

To determine the bacterial and fungal composition of studied kombuchas, the v4 region of the 16S ribosomal RNA (rRNA) gene [16] and the ITS2 region between the 5.8S and LSU rRNA genes [17] were used for sequencing library preparation. For this, 25 ng of extracted microbial gDNA was taken for the amplification stage, and dual indices were used for final libraries and carried out as described in Andreson et al. [2] and Kazantseva et al. [18]. The next-generation sequencing (NGS) was performed using the iSeq100 platform (Illumina, San Diego, CA, USA) with iSeq 100 i1 Reagent v2 and applied as 2 × 150 cycles paired-end protocol for 16S and 300 cycles single-end protocol for ITS2 region. The corresponding bioinformatic data analyses were performed using an open-source BION-meta package [19,20] for bacterial identification and by the Quantitative Insights Into Microbial Ecology [QIIME2, version 2019.10, [21]] for fungal identification, as described in Andreson et al. [2].

#### 2.5.3. Data Quantification

For absolute quantification of the bacterial and yeast species detected by the 16S and ITS rRNA amplicon sequencing in the studied kombuchas, a quantitative real-time PCR (qPCR) on a qTOWER3G thermal cycler (Analytik Jena, Jena, Germany) using calibration curve methodology was performed, as mentioned before Andreson et al. [2]. All experiments were performed in triplicate. Standard primers of amplifying the v4 region of the 16S rRNA gene [16] for bacteria, and the ITS2 region of the ribosomal gene [22] for fungi were used. For cell number calculations, the average genome size of the detected microbial species was taken as 4 Mbp for bacteria and 14 Mbp for yeasts.

To estimate the cell number from qPCR data, the following equation was used:$$Cell\;number=\frac{m\times N_{A}}{GS\times M},$$
where *m* is the mass of DNA (g) measured by qPCR, *N_A_* is the Avogadro constant 6.02 × 10^23^ L/mol, *GS* is the mean genome size in bp, and *M* is the molar mass of 1 bp of dsDNA equal to 660 g/mol.

### 2.6. Data Visualisation

Titratable acid, pH, organic acids, sugars, ethanol, and ITS and 16S amplicon data were visualised using packages “ggplot2” 3.3.6 and “scales” 1.2.0 in R version 4.0.2 22 June 2020 (R Foundation for Statistical Computing, Vienna, Austria).

## 3. Results

### 3.1. LAB-Tailored Kombucha Culture

To modify the original kombucha culture after 7 days of its propagation in the black-tea environment, four lactic-acid bacteria (LAB) species were added to the initial culture. After two weeks of fermentation, the LAB-tailored kombucha was sensorially different from the original drink. While the initial kombucha had a strong acidic taste, the LAB-tailored kombucha was less sour (Appendix A). The sourness was softer and similar to the taste of lactic acid. However, the pH and titratable acidity did not differ significantly between the initial and LAB-tailored fermented drinks.

The bacterial and yeast composition of liquid and SCOBY from the initial and tailored kombucha were determined by 16S or ITS2 metabarcoding, accordingly (Figure 2). In the initial kombucha, the predominant bacterial species was *Komagataeibacter rhaeticus*, whilst *K. europaeus* and *K. intermedius* were identified in smaller proportions. In the LAB-tailored kombucha, *K. rhaeticus* prevailed. While LAB species were detected at the beginning, they were not found later in tailored kombucha by 16S NGS. Additionally, the yeast composition of tailored kombucha differed from the initial drink. The dominant yeast species in the initial kombucha was *Dekkera bruxellensis*, but the tailored kombucha had more than one dominant yeast species. Prevalent yeast species belonged to the genus *Zygosaccharomyces*.

As we could not identify added LAB by 16S metabarcoding in tailored kombucha, we performed qPCR with strain-specific primers at five sampling points during a four-month period (Appendix A). Despite the fact that the measured LAB amount was relatively high at the first sampling point, with values between 10^4^ to 10^5^ cells per millilitre of kombucha, the abundance of LAB species decreased 1–3 log in the second sampling point and stayed below the detection threshold until the end of the fourth month. Still, modified LAB-tailored and propagated kombucha culture was used further in backslopping experiments.

### 3.2. Orange-Juice Kombucha Backslopping

#### 3.2.1. Chemical Composition and Sensory Evaluation

In the preliminary optimisation study, the fermentation of tailored kombucha in orange juice was monitored over a period of five days. It included measurements of pH, titratable acidity, and sensory characterisation (Appendix A). The kombucha fermented in orange juice was sensorially unacceptable after more than two days of fermentation. The fermented beverage was becoming rotten and developing a strong acetic taste. Thus, a single backslopping cycle duration for orange juice was chosen, specifically, two days. The whole experiment continued for five backslopping cycles.

The orange-juice kombucha was evaluated for acidity and metabolites during the whole backslopping process. The pH of natural orange juice was 3.99 ± 0.02 (Figure 3). The highest TA values were detected at the fourth and fifth backslopping cycles, with values of 8.05 ± 0.04 g/L and 7.94 ± 0.06 g/L, respectively. A trend showed that after the end of each backslopping cycle, the TA increased even more than in the previous cycle.

Metabolic analysis showed that despite citric acid being a dominating acid throughout the experiment, its concentration showed minor changes in range of 8.3 to 9.1 g/L (Figure 4A). Acetic acid showed the greatest increase, increasing from 0.04 ± 0.00 to 3.08 ± 0.02 g/L throughout the backslopping cycles (Figure 4A). Ethanol concentration increased throughout the backslopping cycles (Figure 4C). The highest concentration occurred at the end of fermentation, with a value of 23.10 ± 0.04 g/L. The highest sucrose content was detected at the beginning of backslopping, with a value of 40.97 ± 0.09 g/L (Figure 4D). The fructose and glucose concentrations were one-half of the sucrose concentration in the beginning of the fermentation. However, a switch in the consumption of sugars happened between backslopping points 1.2 and 3.1. After that shift, the sucrose concentration was lower than the fructose and glucose concentrations. At the end of the fifth cycle, the sucrose was almost depleted, and some fructose and glucose were left. Organic acid, sugars, and ethanol profiles demonstrated that the kombucha fermentation accelerated during the backslopping, which indicates fastened fermentation after each backslopping cycle (Figure 4E).

By the end of the first cycle, the sensory evaluation showed that the kombucha culture changed orange juice’s taste towards the sour and carbonated it. The fermented orange-juice kombucha had a chemical taste and bitter notes by the end of the second and third cycles, respectively. By the fourth and fifth backslopping cycles, the fermented orange juice was very sour and bitter and had profound chemical notes (Appendix A).

#### 3.2.2. Microbial Composition

The microbial composition of every backslopping cycle was detected solely from SCOBY, as it was the transferring component applied to the next kombucha batch during the backslopping process. The changes in bacterial proportion revealed by 16S metabarcoding and the enumerated data normalised by bacteria-specific qPCR during the backslopping cycle are shown in Figure 5A,B, respectively. The BS 0 cycle signifies the adaptation period of the microorganisms, with the switch between black tea (BS 0.0) and orange juice (BS 0.2) matrices.

The results showed that two main bacterial species—*Komagataeibacter rhaeticus* and *Komagataeibacter intermedius*—were detected throughout the experiment. The *K. rhaeticus* was the only bacteria detected at the point of BS 0.0 where the lowest numbers of bacterial cells were 1.81 ± 0.18 × 10^6^ cells/mL. *K. rhaeticus* prevailed until the point of BS 0.2, when the drink contained the highest number of cells (4.00 ± 0.49 × 10^7^ cells/mL). From the second backslopping cycle (BS 2.2) until the end (BS 5.2), the *K. intermedius* started to dominate over others. The *K. intermedius* cell count was increased to 1.12 ± 0.12 × 10^7^ cells/mL in BS 2.2. At the end of the fifth cycle (BS 5.2), the *K. intermedius* proportion was raised to 0.86, while *K. rhaeticus* was diminished to 0.13, with a lower total cell number of 2.69 ± 0.55 × 10^6^ cells/mL. To some extent, 16S NGS revealed species only at the genus level, so from BS 1.2 to BS 5.2 the proportions of 0.01–0.03 species were identified as members of the genus *Komagataeibacter*. At the end of the experiment on the fermented orange juice, *K. intermedius* was found to be the dominant species. Even though the content of *K. intermedius* increased significantly during the last cycle, the overall number of bacteria showed a decreasing tendency after the point of BS 2.2 and stayed especially low (2.69 ± 0.55 × 10^6^ cells/mL) at the end of the cycles compared to the beginning of the backslopping (downtrend 2.4 log).

The yeasts community was detected by ITS2 metabarcoding along with the cell numbers determined by ITS-specific qPCR in analysed time points, as presented in Figure 5C,D, respectively. The results showed that two genera were detected throughout the experiment. In the beginning, at the point of BS 0.0, the dominant species was *Zygosaccharomyces parabailii*, with a proportion of 0.58, followed by *Schizosaccharomyces pombe* and *Zygosaccharomyces* spp., with proportions of 0.23 and 0.20, respectively. Similarly to the bacterial cell load, the lowest yeast cell count was detected at the time point of BS 0.0, 1.81 ± 0.05 × 10^5^ cells/mL. In the same way, the highest total yeast cell count was measured at the point of BS 0.2, 3.35 ± 0.62 × 10^7^ cells/mL, while species abundances did not change remarkably, with dominating *Z. parabailii* followed by the *Zygosaccharomyces* spp. and *S. pombe*. From BS 2.2 until the end (BS 5.2), the *S. pombe* started to dominate with a proportion of 0.68 and had a cell count of 4.10 ± 0.31 × 10^6^ cells/mL. A minor decrease of dominance occurred at the point of BS 3.2, but in BS 4.2 and BS 5.2, the *S. pombe* content was increased to 0.70 and 0.77, respectively. Therefore, the *S. pombe* dominated at the end of the backslopping cycles. Although, at the end of backslopping, the yeast cell load increased to 1.10 × 10^7^ cells/mL, it is still lower than in the beginning.

### 3.3. Coffee Kombucha Backslopping

#### 3.3.1. Chemical Composition and Sensory Evaluation

The preliminary optimisation study was also conducted in a coffee environment (Appendix A). The fermentation of tailored kombucha was monitored over a period of five days, with measurements taken for pH, titratable acidity, and sensory characteristics. As the sensory properties of fermented drinks became unpleasant after the fourth day, the duration of one backslopping cycle for coffee was limited to four days. The whole experiment continued for five backslopping cycles.

The chemical composition and acidity of kombucha during its backslopping in the coffee infusion were evaluated by analogy to orange juice. The pH of coffee infusion was 5.69 ± 0.04, which was adjusted to a pH of around 4.7–4.9 (Figure 6). The lowest pH obtained during the backslopping process for coffee kombucha was 4.02 ± 0.01. As in the case of orange-juice kombucha, the TA values demonstrated similar changes in acidity during the backslopping cycles. Similar to the orange juice, the later backslopping cycles in coffee ended faster.

Metabolic analysis showed that acetic acid increased from 0.32 ± 0.01 to 4.86 ± 0.02 g/L throughout the backslopping cycles (Figure 7A). The acetic acid concentration rise was higher than corresponding rise in the orange juice. The ethanol concentration increased to 8.58 ± 0.28 g/L at the end of the first cycle but decreased to 5.91 ± 0.48 g/L after the second cycle (Figure 7C). The second and third cycles of backslopping had similar ethanol concentrations. At the end of the fifth backslopping cycle, the concentration of ethanol was increased to 10.23 ± 0.32 g/L. This concentration was two times lower than that in the orange-juice kombucha. The highest sucrose concentration occurred at the beginning of backslopping, with a value of 107.22± 0.01 g/L (Figure 7D), while fructose and glucose were not detected. The sucrose concentration decreased during the backslopping cycles, and the final concentration was 49.06 ± 0.04 g/L. The fructose and glucose amounts increased throughout the backslopping cycles. Coffee kombucha had minor changes in the sugar concentration compared to the orange-juice kombucha, in which the balance of sugars changed at the end of the backslopping cycles. The coffee kombucha organic acid, sugars and ethanol profiles showed the same trends as those of the orange-juice kombucha regarding the acceleration of the fermentation rate during the backslopping (Figure 7E).

The sensory analysis showed that by the fourth day of fermentation, the kombucha culture imparted a sweet and sour taste to the coffee. Additionally, by the end of the first cycle, the coffee kombucha had developed a bitter taste, and by the end of the third cycle, the beverage had acquired carbonisation and a strong bitter taste. In the fourth and fifth backslopping cycles, the coffee kombucha exhibited intense sourness, bitterness, and carbonation (Appendix A).

#### 3.3.2. Microbial Composition

In the coffee kombucha, the microbes were detected from kombucha liquid and SCOBY because both parts were transferred during the backslopping. The changes in bacterial composition during the backslopping cycles in the coffee kombucha detected by metagenomic 16S amplicon NGS and enumerated by bacteria-specific qPCR data are shown in Figure 8A,B, respectively. As in the orange-juice kombucha, the sequencing results revealed that the coffee kombucha contained the same two major bacterial species during the backslopping. *Komagataeibacter rhaeticus* occurred only from point BS 0.0 to BS 1.4 in both parts of the kombucha, and it dominated in the kombucha liquid at all backslopping points (BS 0.0 to BS 5.4). However, the bacterial cell load in the kombucha liquid was relatively low throughout the experiment. On the other hand, the bacterial cell count was quite stable during the whole experiment, reaching, at the endpoint, a value of 1.32 ± 0.07 × 10^5^ cells/mL, which is comparable to that of the backslopping point BS 0.4.

Compared to the SCOBY, the *K. rhaeticus* was not the dominating species at the points of BS 3.4 and BS 4.4. Additionally, the detected bacterial load in SCOBY was twofold higher than in the kombucha liquid. The highest number of bacterial cells in SCOBY was detected for the *K. rhaeticus* at the point of BS 0.4 (1.62 ± 0.23 × 10^7^ cells/mL) and for the *K. intermedius* at the point of BS 3.4 (9.03 ± 0.77 × 10^6^ cells/mL). From the last-mentioned point (BS 3.4), the proportions of both species were approaching equality. By the end of the experiment (BS 5.4), the content levels of the *K. rhaeticus* and *K. intermedius* in the SCOBY were in a ratio of 50:50. Simultaneously, the total count of bacterial cells in SCOBY at the backslopping endpoint (BS 5.4) was 2.25 ± 0.10 × 10^6^ cells/mL, which was approximately sevenfold lower than in the beginning. This result is quite equal to the value measured in the tea medium (BS 0.0). In general, the coffee backslopping showed stabilizing effects on the bacterial composition and content.

The yeast community was detected by metagenomic ITS2 amplicon NGS and is presented with the fungal cell number determined by ITS-specific qPCR in Figure 8C,D, respectively. The results showed that the coffee kombucha contained the same two yeast genera as the orange-juice kombucha. The SCOBY and kombucha liquid had different dominant yeast species in the beginning of the experiment (BS 0.0—black tea). The *Schizosaccharomyces pombe* prevailed in the kombucha liquid, and *Zygosaccharomyces parabailii* in SCOBY. However, the *S. pombe* was more prevalent than was the *Z. parabailii* in the SCOBY at the starting point of the backslopping (BS 0.4—coffee). Throughout the experiment, the *S. pombe* was dominant in the kombucha liquid. Still the total yeast cell count in the kombucha liquid was two- to threefold lower than in the SCOBY, with the lowest value in the liquid at the point of BS 4.4 (2.29 ± 0.65 × 10^2^ cells/mL). At the same time, the SCOBY was a little bit more dynamic. However, the *S. pombe* was beginning to dominate in the SCOBY, but was still less than at the same point in the kombucha liquid. Generally, the yeast population in coffee kombucha liquid and SCOBY decreased during the backslopping experiment, reaching to the values of 6.42 ± 0.26 × 10^5^ cells/mL and 1.18 ± 0.06 × 10^3^ cells/mL, respectively.

## 4. Discussion

Previous research on kombucha has primarily focused on studying its health benefits. Some studies in which kombucha culture was introduced in different environments, such as pomegranate juice [23], grape juice [24], and others [8], accented the description of its antioxidant properties. SCOBY composition is variable depending on its origin and fermentation conditions. Its consortia are mainly enriched by acetic acid bacteria (AAB) and yeasts. This study aimed to develop and characterise new beverages based on modified LAB SCOBY fermentation in non-traditional growth environments by backslopping technology. The focus was on describing the chemical and microbial changes during backslopping cycles that were supported by sensory evaluation.

The first stage of the new kombucha product development was to modify the initial kombucha culture with several LAB species. *Levilactobacillus brevis*, *Lactiplantibacillus plantarum*, *Pediococcus pentosaceus* and *Companilactobacillus paralimentarius* are strains that were isolated from plants [15], and therefore are more capable of adapting to the kombucha environment than are the classical milk-origin LABs. After introducing the LABs to kombucha fermentation, we expected changes in the SCOBY composition, with new probiotic cultures added to it, and the taste modified.

Despite the evident effects on sensory properties, the added LAB did not show measurable viability in kombucha. We could not detect them after serial backslopping cycles even by strain-specific primers by qPCR. Probably, these strains could not compete for substrates due to the microbial richness in the kombucha. However, the composition of original microbial consortia changed after LAB tailoring, shifting from three species (*K. intermedius*, *K. rhaeticus*, and *K. europeaus*) in the initial composition, to the one for the LAB-tailored kombucha. Despite the fact that we detected lactic acid throughout the entire fermentation process, it was not dominating the organic acid. In their study, Nguyen et al. [25,26] supplemented kombucha with LAB strains and demonstrated a significant increase in glucuronic acid and antioxidant activity However, they did not measure the abundance of the lactic acid bacteria after kombucha fermentation. Our data also proved that, although the tailored kombucha had a low abundance of LAB strains, it contained a higher amount of gluconic acid than did the initial kombucha (Appendix A). Thus, we can assume that the added LAB influenced kombucha properties positively.

In this study, we observed dynamic changes in microbial composition during the backslopping cycles. In both environments, initially detected bacterial species disappeared through several backslopping cycles, but they were again detected at the end of the experiment. Thus, it showed fluctuation in species’ behaviour and their dependence on the environment. Moreover, the dominant species were different for various matrices. In the initial tea kombucha, the prevailing bacterium was *K. rhaeticus*, but in the orange juice, it was *K. intermedius*; whereas, in the coffee kombucha, the *K. rhaeticus* retained its dominance only by the end of the backslopping. From the perspective of yeasts, the LAB addition totally changed the yeast composition of kombucha. In the initial kombucha, the primary yeast was *Dekkera bruxellensis*, but the *Zygosccharomyces* species dominated in the LAB-tailored kombucha. The yeast composition in the orange juice and coffee did not change back to the initial kombucha composition, as had happened with bacteria. These changes demonstrate that the microorganisms which are detectable in the initial kombucha do not disappear, but are still present in very small amounts and are metabolically active, waiting for the environment to become favourable to increase their populations. Thus, backslopping plays an important role in the balance of the microbial consortia, supporting or reducing the growth of definite microorganisms, depending on the environment.

Although, in both environments, backslopping did not significantly affect the chemical properties of kombucha, the results showed relatively high concentrations of ethanol during the analysed process. The ethanol concentrations reached 23.10 g/L and 10.23 g/L in orange juice and coffee environments, respectively. Chen and Liu [27] also showed that the ethanol concentration in kombucha increased with time and reached 5.5 g/L by the end of fermentation. Another study by Chakravorty et al. [12] determined the highest concentration of ethanol to be on day 7, with a value of 0.28 g/L. Besides, Tan et al. [28] showed that ethanol concentration reached 70.7 g/L in juice fermented with yeast. It is known that different species have different levels of ethanol tolerance [3,29]. Thus, we assume that the elevated concentrations of ethanol in the coffee and orange-juice kombuchas affected the viability of bacteria and yeast. The lowest bacterial cell counts were detected when the ethanol concentration was the highest in both orange juice and coffee. However, the yeast cell counts were not the lowest in orange juice at that point compared to that in coffee.

One additional trend observed in coffee kombucha was that an increase in ethanol levels contributed to a higher concentration of acetic acid. This observation aligns with the classical kombucha metabolism, in which the yeast produces ethanol, which is then converted to acetic and other acids by AAB [5,6]. However, this tendency was not as apparent in orange-juice kombucha. Metagenomic results indicated that the content of *S. pombe* was initially high, slightly decreased, and by the end of the backslopping process, the yeast content was even higher than at the beginning of the experiment. This may be attributed to the yeast species adapting to the orange juice environment. Additionally, it appears that the ethanol concentration in both orange juice and coffee kombucha reached levels deemed unacceptable, preventing the AAB from converting it to acetic acid. The *Komagataeibacter* species exhibited tolerance to the elevated ethanol environment and adapted accordingly, whereas the supplemented LAB species did not.

Despite the fact that the orange-juice kombucha acquired a carbonated orange lemonade-like taste after two days of fermentation, prolonging the backslopping cycles led to changes in its sensory characteristics, rendering the taste sensorially unacceptable. On the other hand, coffee kombucha exhibited a pleasant balance of sweet and sour flavours and developed some carbonation after four days of fermentation. However, an excessive number of backslopping cycles caused a deterioration in its taste. In general, the backslopping experiments conducted in both environments demonstrated accelerated fermentation and a shortened backslopping cycle. Therefore, when it comes to production purposes, the application of backslopping technology for nontraditional kombucha matrices should be evaluated individually for each specific environment and condition.

Our sensory evaluation highlighted the importance of thorough monitoring of all chemical and microbial changes as a critical aspect of developing a new beverage. The study revealed that each backslopping cycle reduced the time required for bacterial adaptation, resulting in an accelerated fermentation process and earlier attainment of acceptable taste and odour. These changes must be taken into account to achieve a stable product with consistent sensory and chemical qualities during production. Moreover, meticulous monitoring of the ethanol concentration is necessary to market the beverage as a soft drink [30]. In terms of enhancing the health benefits of kombucha, it is desirable to enrich it with additional beneficial microbes, such as LAB, which positively impact the beverage’s chemical properties, sensory characteristics, and stability [31]. It is possible to find a SCOBY that already contains LAB [2] or select LAB strains that can adapt to the existing SCOBY culture.

## 5. Conclusions

The development of novel fermented drinks with unique flavours and properties poses a significant challenge. The SCOBY used in kombucha production is a complex and wild culture that dynamically adapts to new conditions and environment. This adaptability makes it difficult to achieve consistent batch-to-batch quality and control in the industrial manufacturing of kombucha. In our study, we examined the changes in the chemical and microbiological compositions of kombucha tailored by LAB cultures in orange juice and coffee, using the backslopping technology. Although the added lactic acid bacteria were not detected in the kombucha, the LAB-tailored kombucha exhibited improved sensory parameters and a modified microbial composition compared to the initial beverage. The transfer of the kombucha culture to the new matrices changed the original microbial community, and subsequent backslopping cycles further modified the consortia composition, thereby accelerating the fermentation rate.

There are various industrial technologies available for producing new kombucha-based beverages. The first option is to utilise the classical kombucha liquid and SCOBY and introduce them into the new environment. This approach requires more initial kombucha volume but tends to result in a more stable product quality. The second option is to employ the backslopping technology, which is easier to implement, but requires diligent and continuous monitoring of the microbial consortia and metabolites changes. Our study elucidated the advantages and disadvantages of different technologies for producing new kombucha-based beverages. Depending on the production capacity and available options, kombucha brewers can select the most optimal technology and adapt it to their specific goals.

## Figures and Tables

**Figure 1 foods-12-03545-f001:**
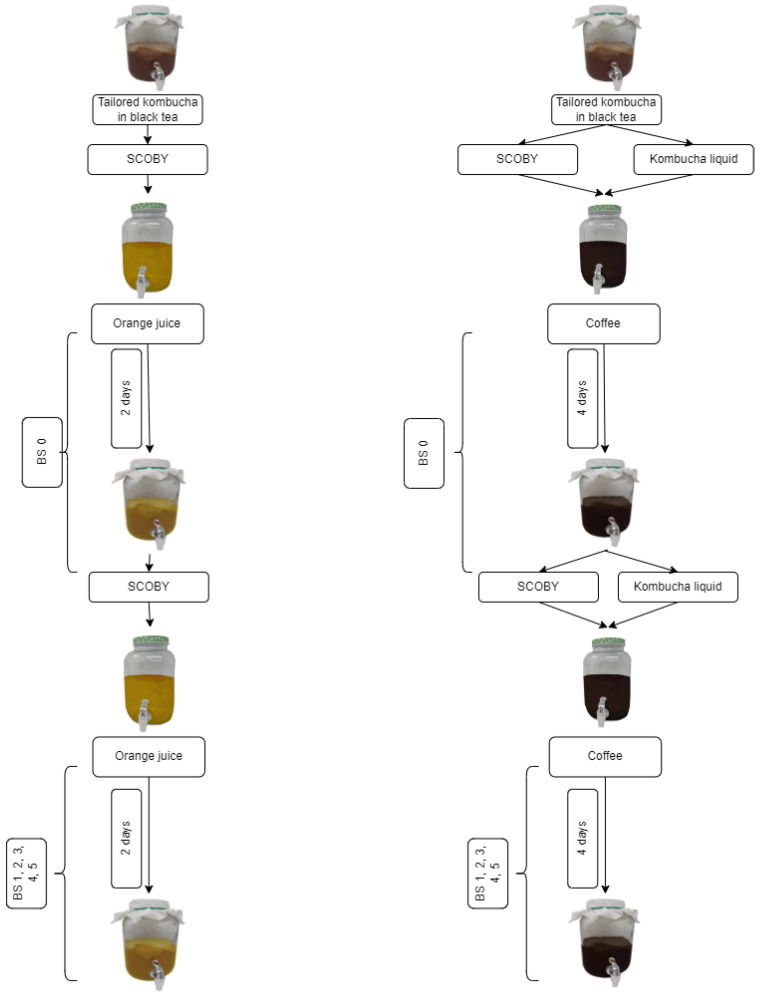
The schematic figure of the backslopping (BS) experiment workflow. The samples for pH, TA, organic acid, and sugar measurements were collected and analysed, and sensory analysis was conducted every day throughout the experiment. Samples for metagenomic analyses were collected and analysed at the end of each backslopping cycle.

**Figure 2 foods-12-03545-f002:**
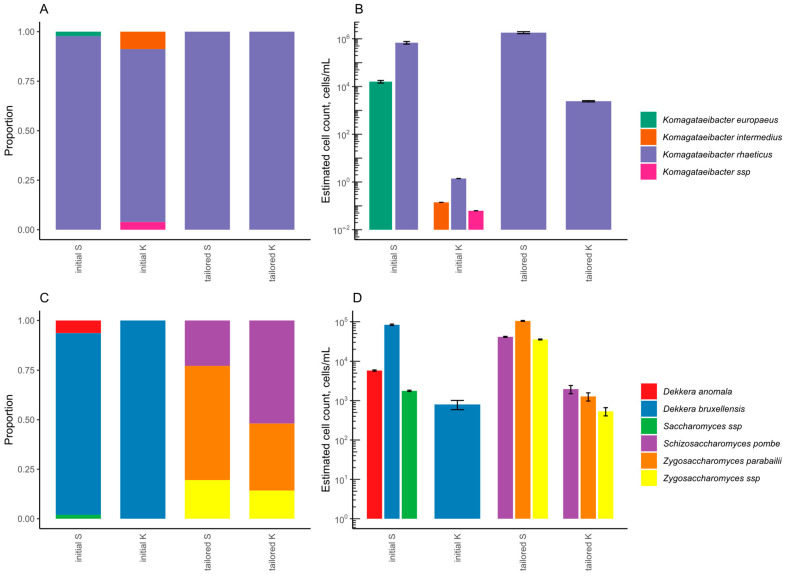
Microbial composition of initial and tailored kombucha at the end of the backslopping process according to 16S NGS for bacteria (**A**,**B**) and ITS2 NGS for yeast (**C**,**D**), graphed as relative (**A**,**C**) and normalised abundances (**B**,**D**). The abbreviation S stands for SCOBY, and K for kombucha liquid.

**Figure 3 foods-12-03545-f003:**
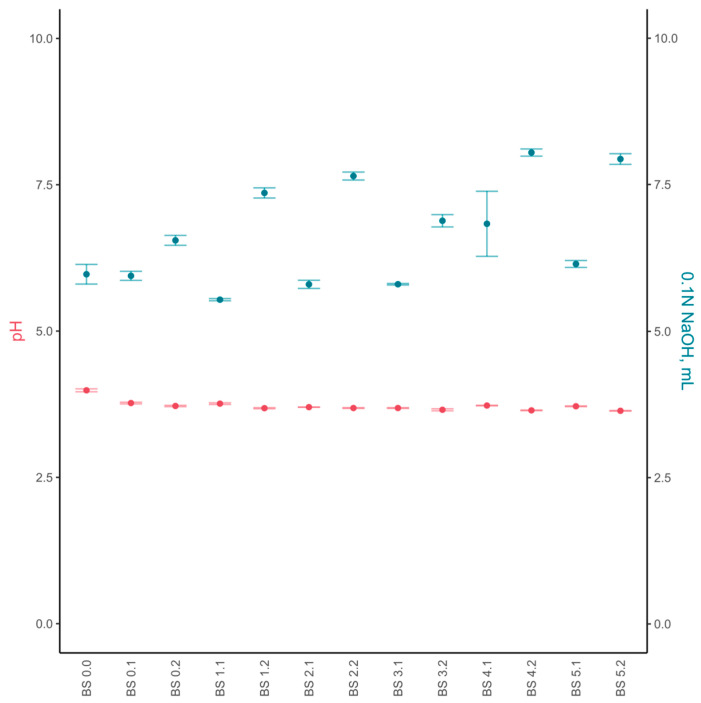
The pH (red dots) and titratable acidity (blue dots) values of orange-juice kombucha during backslopping cycles. Titratable acidity units are mL of 0.1 N NaOH. The abbreviation BS stands for backslopping. The first number shows the backslopping cycle and the second number refers to the day of the backslopping cycle.

**Figure 4 foods-12-03545-f004:**
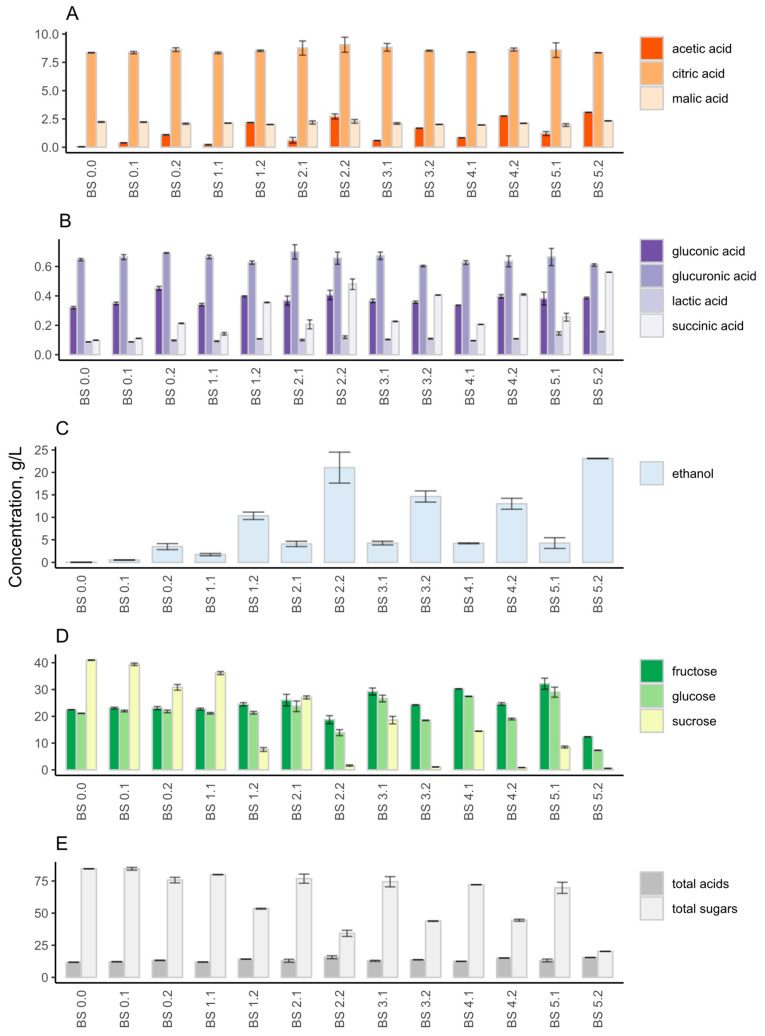
Organic acids, sugars, and ethanol concentrations (g/L) of orange-juice kombucha during backslopping. Concentrations of acetic, citric, and malic acid (**A**); gluconic, glucuronic, lactic, and succinic acid (**B**); ethanol (**C**); fructose, glucose, and sucrose (**D**); and total acids and sugars (**E**) are indicated. The abbreviation BS stands for backslopping. The first number shows the backslopping cycle and the second number states the day of the backslopping cycle.

**Figure 5 foods-12-03545-f005:**
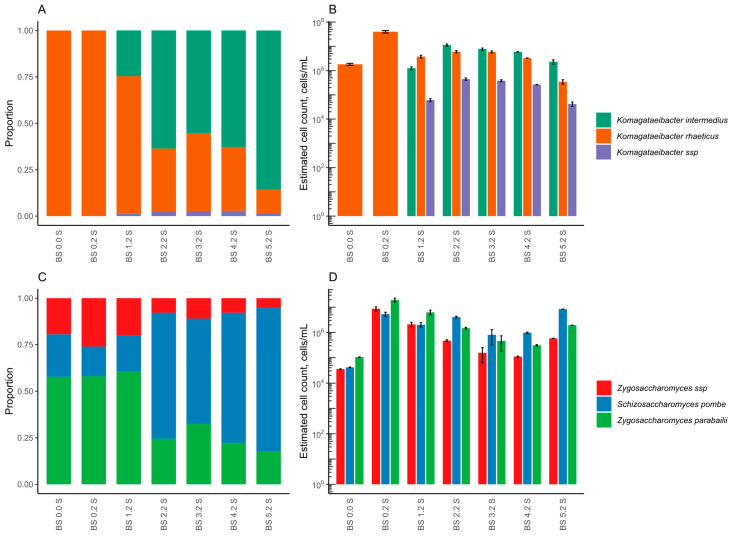
Microbial composition during orange juice backslopping cycles according to 16S NGS for bacteria (**A**,**B**) and ITS2 NGS for yeast (**C**,**D**), graphed as relative (**A**,**C**) and normalised abundances (**B**,**D**). The abbreviation BS stands for backslopping. The first number shows the backslopping cycle and the second number states the day of the backslopping cycle.

**Figure 6 foods-12-03545-f006:**
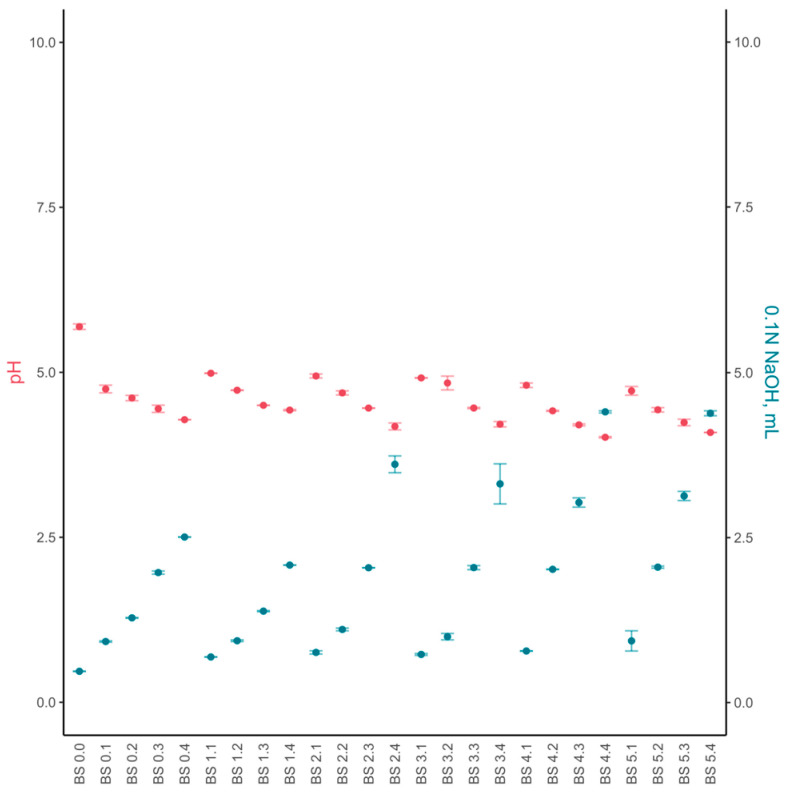
The pH (red dots) and titratable acidity (blue dots) values of coffee kombucha during backslopping cycles. Titratable acidity units are mL of 0.1 N NaOH. The abbreviation BS stands for backslopping. The first number shows the backslopping cycle and the second number states the day of the backslopping cycle.

**Figure 7 foods-12-03545-f007:**
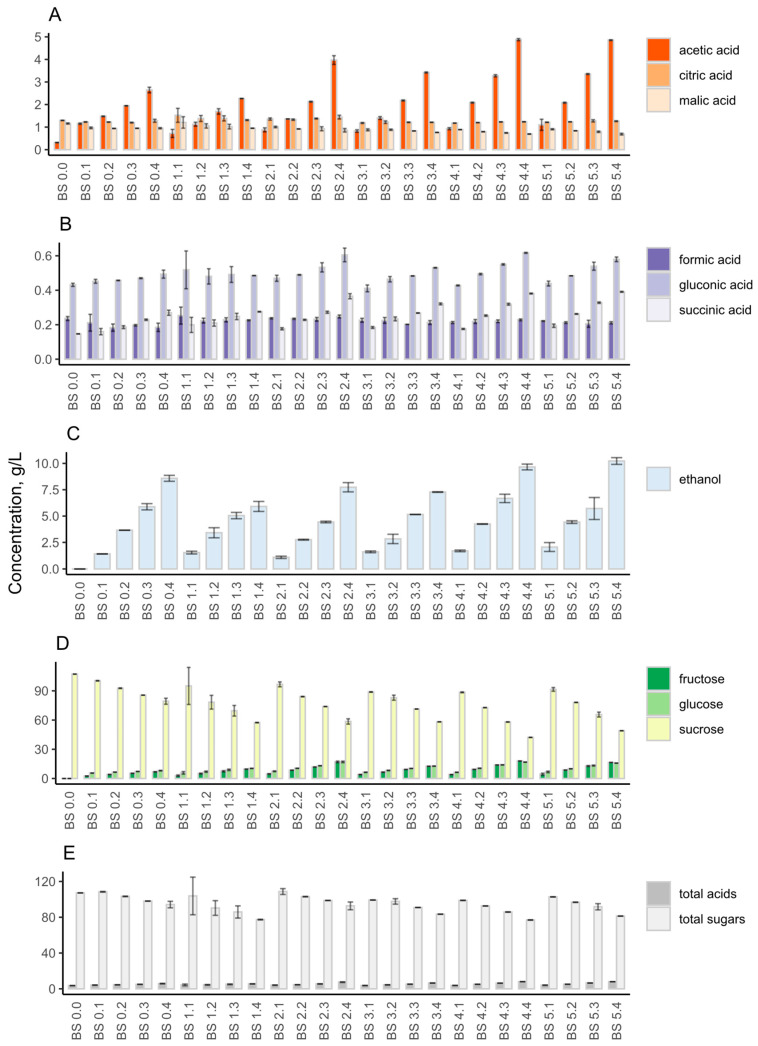
Organic acids, sugars, and ethanol concentrations (g/L) of coffee kombucha during backslopping. Concentration of acetic, citric, and malic acid (**A**); formic, gluconic, and succinic acid (**B**); ethanol (**C**); fructose, glucose, and sucrose (**D**) and total acids and sugars (**E**) are indicated. The abbreviation BS stands for backslopping. The first number shows the backslopping cycle and the second number states the day of the backslopping cycle.

**Figure 8 foods-12-03545-f008:**
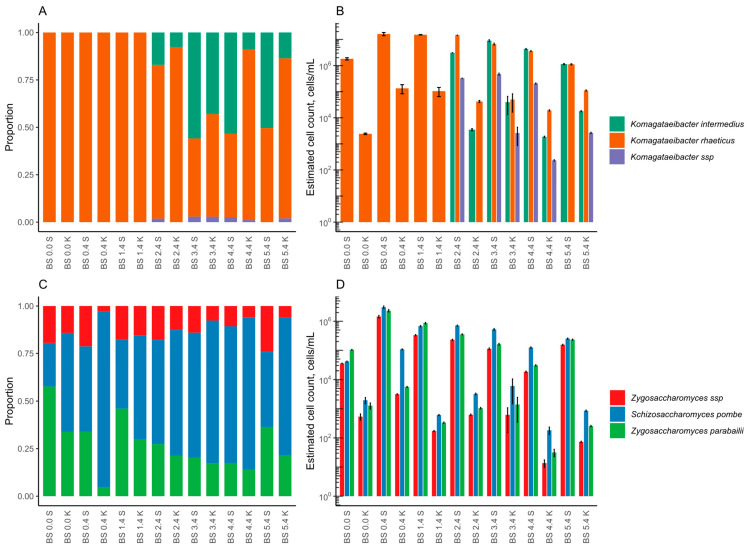
Microbial composition during coffee backslopping cycles according to 16S NGS for bacteria (**A**,**B**) and ITS2 NGS for yeast (**C**,**D**), graphed as relative (**A**,**C**) and normalised abundances (**B**,**D**). The abbreviation BS stands for backslopping. The first number shows the backslopping cycle and the second number states the day of the backslopping cycle.

## Data Availability

The data used to support the findings of this study can be made available by the corresponding author upon request.

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
