# Peer review of "Evaluation of Microbial Dynamics of Kombucha Consortia upon Continuous Backslopping in Coffee and Orange Juice"

_foods, 2023, doi:10.3390/foods12193545_

Round 1

Reviewer 1 Report

This is an interesting and well-written paper, reporting and discussing the results of the study that evaluated the changes in microbiota, chemical properties, and sensory of kumbucha during the process of preparation with less conventional matrices (orange juice and coffee), while using backslopping technology. The effect of adding LAB multi-strain at the initial kumbucha culture was also evaluated in this experimental design of the study (LAB-tailored kombucha fermentation). The authors concluded that the LAB-tailored kombucha improved sensory parameters and modified the microbial composition of the fermented product, and the use of backslopping technology accelerated the fermentation rate. Despite the quality of the manuscript, there are minor clarification and precision.

• Section 2.1 Creation of kombucha culture         

The concentration rate of added LAB strains was 1x10^7 cells/mL, which seems low, and usually expressed in CFU per amount of sample used. Was cells/mL the right unit in this case?      

• Section 2.3 Determination of pH and titrable acidity

Please precise the number of measurement replicates.

• Section 3.1 LAB-tailored kombucha culture

The authors did not find significant differences between the initial and LAB-tailored fermented drinks in terms of pH and titrable acidity (TA). However, the TA values were 2,7 vs. 4,2, respectively. Is this difference statistically insignificant?

• Subsection 3.2.1 Chemical composition and sensory evaluation

The TA value was superior at the fourth backslopping cycle (9,05 g/L) than at the fifth one (7,94 g/L). Is this result coherent to the trend announced in the next sentence?

• Section 3.2.2 (3rd §)

The term "yeast community" is more convenient than "yeast diversity" since only two genera were described for the product. 

Reviewer 2 Report

The study studied the chemical properties and microbial growth dynamics of lactic acid bacteria-tailored (LAB tailored) kombucha culture by 16S rRNA next-generation sequencing in coffee and orange juice during a back-slopping process that spanned five cycles, each lasting two to four days. The back slopping changed the culture composition and accelerated the fermentation. This study also gives an overview of pros and cons of back slopping technology for the production of kombucha-based beverages. This study interesting, however there are some points that need to be clarified. Recommendations were published after the revision.

1.Keywords in ‘lactic acid bacteria’ and “fermented beverage”, change to Coffee “kombucha backslopping” and “Orange juice kombucha backslopping” is better.

2. Whether the "Creation of kombucha culture" in the experimental method has literature evidence?

3. The meaning of“a piece of SCOBY” in method 2.2 and how to add it should be explained.

4.“(Espinosa-Gongora et al., 2016; McDonald et al., 2016; www.box.com/bion) ” should appear in the reference, and only the number of the reference should be marked in the article.

5.6.02×1023 1/mol” should be corrected to “6.02×1023 L/mol.”

6. Where is the table that the full text has been mentioning?

7.2.29 × ± 0.65 102 cells/mL” should be corrected to “2.29 ± 0.65 ×102 cells/mL.”

8. The phenolic acid content and antioxidative activity should be analyzed (Food Chemistry, 402(2023): 134231.).

9. Changes in the content of some acids measured in Figures 4B and 7B are not mentioned in the results.

10. The metabolites changes should be analyzed (International Microbiology, 25 (3): 417-426.).

11. In the conclusion, the results of this article are less mentioned

12. The article lacks specific elaboration on the advantages and disadvantages of different technologies for producing new kombucha-based beverages.

The study studied the chemical properties and microbial growth dynamics of lactic acid bacteria-tailored (LAB tailored) kombucha culture by 16S rRNA next-generation sequencing in coffee and orange juice during a back-slopping process that spanned five cycles, each lasting two to four days. The back slopping changed the culture composition and accelerated the fermentation. This study also gives an overview of pros and cons of back slopping technology for the production of kombucha-based beverages. This study interesting, however there are some points that need to be clarified.

1.Keywords in ‘lactic acid bacteria’ and “fermented beverage”, change to Coffee “kombucha backslopping” and “Orange juice kombucha backslopping” is better.

2. Whether the "Creation of kombucha culture" in the experimental method has literature evidence?

3. The meaning of“a piece of SCOBY” in method 2.2 and how to add it should be explained.

4.“(Espinosa-Gongora et al., 2016; McDonald et al., 2016; www.box.com/bion) ” should appear in the reference, and only the number of the reference should be marked in the article.

5.6.02×1023 1/mol” should be corrected to “6.02×1023 L/mol.”

6. Where is the table that the full text has been mentioning?

7.2.29 × ± 0.65 102 cells/mL” should be corrected to “2.29 ± 0.65 ×102 cells/mL.”

8. The phenolic acid content and antioxidative activity should be analyzed (Food Chemistry, 402(2023): 134231.).

9. Changes in the content of some acids measured in Figures 4B and 7B are not mentioned in the results.

10. The metabolites changes should be analyzed (International Microbiology, 25 (3): 417-426.).

11. In the conclusion, the results of this article are less mentioned

12. The article lacks specific elaboration on the advantages and disadvantages of different technologies for producing new kombucha-based beverages.

Round 2

Reviewer 2 Report

1. The phenolic acid content and antioxidative activity should be analyzed, especially after fermentation. Please refer to this reference (Food Chemistry, 402(2023): 134231.).

2. The phenolic acid should be analyzed by HPLC. please refer this reference(International Journal of Food Science and Technology, 2020, 55(6): 2531-2540.).

Please update the reference in recent years.

1. The phenolic acid content and antioxidative activity should be analyzed, especially after fermentation. Please refer to this reference (Food Chemistry, 402(2023): 134231.).

2. The phenolic acid should be analyzed by HPLC. please refer this reference(International Journal of Food Science and Technology, 2020, 55(6): 2531-2540.).

Please update the reference in recent years.

Author Response

Dear Reviewer 2,

Your referred to the articles Food Chemistry, 402(2023): 134231 titled The positive correlation of antioxidant activity and prebiotic effect about oat phenolic compounds and International Journal of Food Science and Technology, 2020, 55(6): 2531-2540 titled In vitro evaluation of digestive enzyme inhibition and antioxidant effects of naked oat phenolic acid compound (OPC). Both articles were done by the same research group where the focus of their study was to evaluate the effect of phenolic compounds and antioxidative activity in oats.

However, the scope of our work was to evaluate the microbial dynamics of kombucha consortia upon continuous backslopping in coffee and orange juice. We understand that both phenolic acid and antioxidant activity are important for fermentation and the health impact of kombucha, but we want to stress one more time it was not the focus of our research. Kombucha is a fermented drink rich and diverse in various metabolites. Although, some metabolites like organic acids were evaluated during this study, then these small molecules have a supportive role for this manuscript and would help to understand the changes of sensorial profile. We appreciate your advice and suggestions, but it would broaden the manuscript topic even more and takes the focus away from the consortium evaluation studies.

Best regards,

Maret Andreson